# Towards Robust Robot Control in Cartesian Space Using an Infrastructureless Head- and Eye-Gaze Interface

**DOI:** 10.3390/s21051798

**Published:** 2021-03-05

**Authors:** Lukas Wöhle, Marion Gebhard

**Affiliations:** Group of Sensors and Actuators, Department of Electrical Engineering and Applied Sciences, Westphalian University of Applied Sciences, 45877 Gelsenkirchen, Germany; marion.gebhard@w-hs.de

**Keywords:** data fusion, MARG-sensors, hands-free interface, pose estimation, human robot collaboration, robot control in cartesian space, multisensory interface, gaze control

## Abstract

This paper presents a lightweight, infrastructureless head-worn interface for robust and real-time robot control in Cartesian space using head- and eye-gaze. The interface comes at a total weight of just 162 g. It combines a state-of-the-art visual simultaneous localization and mapping algorithm (ORB-SLAM 2) for RGB-D cameras with a Magnetic Angular rate Gravity (MARG)-sensor filter. The data fusion process is designed to dynamically switch between magnetic, inertial and visual heading sources to enable robust orientation estimation under various disturbances, e.g., magnetic disturbances or degraded visual sensor data. The interface furthermore delivers accurate eye- and head-gaze vectors to enable precise robot end effector (EFF) positioning and employs a head motion mapping technique to effectively control the robots end effector orientation. An experimental proof of concept demonstrates that the proposed interface and its data fusion process generate reliable and robust pose estimation. The three-dimensional head- and eye-gaze position estimation pipeline delivers a mean Euclidean error of 19.0±15.7 mm for head-gaze and 27.4±21.8 mm for eye-gaze at a distance of 0.3–1.1 m to the user. This indicates that the proposed interface offers a precise control mechanism for hands-free and full six degree of freedom (DoF) robot teleoperation in Cartesian space by head- or eye-gaze and head motion.

## 1. Introduction

Direct human robot collaboration demands robust interfaces to interact with or control a robotic system in a human safe manner. Especially in situations where the hands of a person are either occupied (industry 4.0) or not usable, e.g., people with severe physical disabilities, a safe, reliable and intuitive communication source that enables a direct interaction with the robotic system should be provided. In the context of assistive robotics, most of these interfaces have been designed hands-free [1,2,3].

A robust hands-free interface provides the opportunity for people suffering from physical motor impairments the opportunity to be (re)integrated into working life, e.g., for pick and place tasks in a library workplace designed for people with tetraplegia [4]. The question of how an interface could be designed to effectively and intuitively allow for hands-free robot control has drawn significant research attention in the last decade [5,6]. Recent approaches focus on the use of head motion or eye-gaze tracking data to allow for direct robot control since both modalities are naturally correlated with direct interaction intention and enable accurate control mechanisms [7]. Gaze based control signals can accelerate and simplify human robot collaboration, especially when it comes to object targeting in pick and place tasks which is essential in the context of human robot collaboration [2,8]. We divide gaze into two subcategories based on the modality used to extract the gaze vector. The most known gaze vector is eye-gaze. Eye-gaze vectors represent the three-dimensional vector from the humans eye to a target gaze point. The direction of the vector changes based on the eye motion [9]. Head-gaze on the other hand describes a vector that is orthogonal with respect to the rotation axis of the humans head and perpendicular to the face. The direction of the vector depends on the orientation of the head [10], see Figure 1b for a depiction of gaze vectors. Accuracy, affordability and mobility are key factors for eye or head-gaze based interfaces to enable intuitive human robot collaboration and to transfer research and development to further applications and end users.

This paper presents a lightweight, infrastructureless head-worn interface for robust and real-time robot control with a total weight of just 162 g, see Figure 1a. It allows for head motion, head-gaze and eye-gaze based robot teleoperation in six degrees of freedom (DoF), three DoF translation and rotation, respectively. The interface combines a head-worn eye tracker, an RGB-D world camera and a custom MARG-sensor board to calculate the users head pose and a 3D gaze point which is the input target point for the robotic end effector in Cartesian position coordinates. The interface furthermore enables orientation control of the end effector (EFF) by using direct head motion mapping based control.

## 2. State of the Art Head and Eye-Gaze Interfaces

Recent Interfaces focus on the use of head motion or eye tracking data to allow for continuous robot control and deliver a low-cost and intuitive control mechanism [6,11].

Head motion based interfaces usually employ low-cost sensors, i.e., Magnetic AngularRate Gravity (MARG)-sensors, to estimate orientation without a need for static infrastructure, e.g., markers, that would limit the useable motion range and environment [1]. The orientation estimation from these sensors is based on the angular rate integration measured by the gyroscope. This raw signal suffers from various noise terms, especially gyroscope offset, that result in drift of the orientation estimation and therefore reduces accuracy. The drift is usually compensated by using global reference vector measurements from accelerometer and magnetometer [12,13]. Whether a magnetometer can be used to correct for heading errors depends on the magnetic environment [14]. Since the magnetic environment for robotic collaboration is at most unstable in indoor environments and nearby the robot, functional safety cannot be guaranteed when using MARG-sensors only. Some industrial-grade commercially available MARG-sensor systems offer strategies to enable a robust orientation estimation in magnetically challenging environments, e.g., XSens MTi series. This MARG-sensor has a build in feature called active heading stabilization (AHS) to enable a low drift unreferenced yaw orientation estimation. This feature is able to deliver yaw drift as low as 1–5 degrees per hour depending on the application [15]. Besides orientation, MARG-sensors can be used to estimate position, at least for a certain period of time. Centimeter accurate position estimation with MARG-sensors however is typically based on data fusion with external sensors (e.g., GPS) [15]. In GPS denied environments (indoor applications) dead reckoning based on MARG-sensors only can be used. This method relies on double integration of the acceleration to extract velocity and position. The double integration step will accumulate every minuscule error and position accuracy decreases rapidly. Depending on the desired application, MARG-sensor only orientation and position estimation accuracy’s might be sufficient. In a human robot collaboration scenario these errors should ideally be removed completely or be kept at a minimum. This is especially true when it comes to 3D gaze point estimation in Cartesian space. A precise position estimation must be proved to calculate an accurate gaze position in Cartesian space. Every head position error will directly influence the gaze point prediction. Therefore, other heading reference sources should be provided to account for the orientation and position drift. Furthermore, using head motion only comes at the cost of having to switch between various motion groups to map from underrepresented 3D motion space of the head to the full 6D end effector motion range.

Eye-gaze based robot control utilizes the natural eye motion of a human. A recent approach utilizes 2D gaze points and gaze gestures to control the pose of a robotic arm in three-dimensional space [11]. The user can switch between motion groups by gaze gestures to control the arm in various directions (xy-plane, z-plane, rotational xy, rotational z) and control the robot by gazing at a dynamic command area (ROI) at the center of the robots’ gripper. This interface needs to track the robots EFF position in order to specify the dynamic command areas and therefore needs a fiducial marker at the EFF, so it relies on infrastructure in terms of stationary markers in the real world scenario. Furthermore, the interface needs more motion group transitions since it only generates two-dimensional commands for a six dimensional control problem.

Tostado et al. [16] proposed an interface decoding eye-gaze into a position in 3D space for robotic end point control. The proposed system is capable of controlling a robotic end effector in 3D space by directly gazing at a specific object or point. The system consists of two stationary eye cameras in front of the user and thus is dependent on the infrastructure which limits the possible workspace related to camera field of view and motion range. Furthermore, the system does not include head tracking capabilities and therefore assumes a fixed head position, which is ensured by a chin rest, further reducing mobility.

Scalera et al. [17] present a robotic system that enables robotic painting by eye-gaze. The work addresses the prior mentioned mobility restrictions by using a remote eye tracking bar (Tobii Eye Tracker 4C) which enables accurate eye-gaze tracking on a computer screen and accounts for head motions of the user. The filtered eye-gaze coordinates on the computer screen are the input coordinates for the TCP position on the canvas. This approach eliminates most mobility restrictions but relies on the stationary eye-tracking camera attached to a computer screen and only supports two-dimensional commands in a single plane from the computer screen mapped onto the robots’ workspace.

Dezmien et al. [18] developed an interface for eye-gaze based robotic writing and drawing. Similarly to Scalera et al. this work utilizes a remote eye tracking bar (Tobii Eye X) to track eye gaze points of a user on a 2D canvas to directly control a robotic arm drawing on the canvas. The approach uses the Tobii Eye X’s head tracking capability to attach or detach the pen on the canvas. Likewise to the prior mentioned approach, the interface enables direct low cost eye-gaze robot control but relies on the stationary camera hardware and is applicable only in a two-dimensional plane.

A recent approach delivers a potential interface for a full eye-gaze based control of a robotic arm by combining eye tracking, head tracking and depth information [2]. This interface however is dependent on a stationary infrared motion tracking system for head pose estimation. The motion capture system cannot be used in mobile applications and furthermore exceeds reasonable costs for a control interface. The control approach only includes three-dimensional end effector position control and does not include end effector orientation. The control approach utilizes the human operators hand as the end effector rather than a robotic gripper. The operators hand is coupled to the robotic arm by a magnetic coupling and therefore can only be used by people that are able to close their hands and grab an object.

The HoloLens 1 is a commercially available interface that is capable of delivering accurate three-dimensional head-gaze but lacks the ability to track eye positions and deliver eye-gaze. The HoloLens one is weighing 579 g in total [19]. The center of mass is at the front of the head and might therefore be too heavy for long uses from people with severe physical disabilities. The new generation of the HoloLens, the HoloLens 2, is able to deliver eye and head-gaze vectors and comes at a total weight of 566 g distributed more equally [20]. To the best of our knowledge, it has yet to be researched if the HoloLens 2 is more suitable for people with severe physical disabilities in long term use.

With the recent technological advantages in camera miniaturization, mobile eye tracking and computer vision this work aims to fill the gap and propose an infrastructureless and lightweight solution for accurate head- and eye-gaze based six DoF robot control in Cartesian space to facilitate hands free and multi-modal human robot collaboration. It enables head motion, head-gaze and eye-gaze based robot teleoperation in six degrees of freedom. The interface combines a binocular head-worn eye tracker, a low-cost RGB-D world camera and a low-cost custom MARG-sensor board to calculate the users head pose and gaze point in three-dimensional space with respect to a user defined world coordinate system. The proposed system does not rely on stationary cameras and is therefore infrastructureless and mobile regarding potential operational environments and usable motion space. The interface utilizes a state of the art and completely open source visual SLAM (simultaneous localization and mapping) approach, namely ORB-SLAM 2 [21] and fuses it with the orientation measurements of a MARG-sensor board to estimate an accurate head pose. The proposed data fusion enables a robust orientation estimation, even when visual data is lost for a large period of time (≥25 s). For eye-gaze control, a lightweight binocular eye tracker is used to extract a 3D gaze point from the RGB-D cameras depth image which is transformed into the coordinate system defined by the estimated head pose. Head-gaze control is achieved by using the depth cameras center pixel as the gaze vector origin. The three-dimensional gaze point is used as the input control for the robotic end effector. Another feature of the interface is switching from Cartesian position to Cartesian orientation control by a simple eye blink. The end effectors’ orientation control is based on the head motion mapping presented in [1]. An eye safe laser embedded into the interface shows the head-gaze vectors endpoint for direct user feedback.

## 3. Head- and Eye-Gaze Interface

Within this work an interface for six dimensional robot teleoperation in Cartesian space, with three dimensions for orientation and three dimensions for position control, respectively, is proposed. This interface is capable of head- or eye-gaze based point-to-point position teleoperation of a robotic arm as well as head motion based EFF orientation control. This is achieved by combining an active infrared stereo camera and a custom MARG-sensor in a 3D printed case that is mounted on top of a mobile binocular eye tracker. The interface is independent of external infrastructure and therefore usable within most indoor environments.

### 3.1. Interface System Setup

The Interface consists of three main hardware parts. The infrared stereo camera, a custom MARG-sensor board and a binocular mobile eye tracker with a USB-C world camera connector. This hardware setup tackles various software tasks which rely on each other and enable the robot teleoperation. We divide these tasks into the following categories and explain them in detail in the subsections below: (A) 3D head pose estimation through visual and inertial data fusion, (B) 3D gaze estimation with respect to the robot coordinate frame and (C) the application interface for the robot control. To align all coordinate frames the head pose estimation block also features an input for any absolute pose measurement system (e.g., fiducial marker based pose estimation, infrared marker pose estimations or a three point coordinate system alignment procedure). All software components besides the MARG-filter framework are written in C++ and embedded in the Robot Operating System (ROS) [22]. ROS enables fast and reliable inter device and software networking. The framework already delivers a lot of software packages to interface various robots, vision systems and much other hardware which enables fast integration and merging of the proposed system into various applications. Figure 2 depicts a general overview of the proposed Interface and its associated hard- and software components.

### 3.2. Interface Hardware Setup

In this work we use an active infrared stereo camera, namely the d435 from Intel^®^ RealSense™ [23]. This camera uses stereo image rectification to estimate depth. Furthermore, it offers an infrared image projector to illuminate the scene for dense depth estimation at varying lighting conditions. The camera provides calibrated and rectified infrared images, depth image and RGB image at up to 90 frames per second depending on the image size. The camera is attached to a custom 3D printed camera mount with tabs for a headband. The camera mount also features an encapsulation for a custom MARG-sensor board. The MARG-sensor system features an Espressif 32 bit dual-core microcontroller unit (MCU) as well as a low power 9-axis ICM 20948 InvenSense MARG-sensor. The MCU is running the FreeRTOS real-time operating system on both cores at 1 kHz scheduler tick rate. The MCU publishes and receives data at 100 Hz via micro-USB over full-duplex UART [24]. One of the GPIO’s is directly soldered to an eye-safe m1 visible red laser, which is centered above the first d435 cameras infrared image sensor. This laser is dot is used for direct user feedback but is not involved in the position or orientation estimation. This custom sensor mount is resting on a binocular eye tracker frame. We use the open source pupil core binocular c-mount eye tracker for this purpose [25]. The d435 is connected to the USB-C plug of the eye tracker. The camera mount has a notch that is placed over the glasses frame and can be secured via a headband on the users head. This stabilizes the camera with respect to the eye tracker and distributes forces from the nose rest of the tracker to the complete circumference of the head therefore reducing slippage from the eye tracker and increases wearing comfort. The complete Interface weighs 162 g. Figure 1a depicts the designed Interface. The interface USB cables are plugged into a standard laptop computer that is running the various C++ nodes on the ROS software stack, compare Figure 2. The laptop is equipped with a 64 Bit i5-5941 quad-core CPU and 6 GB of RAM running Ubuntu 16.04 with ROS version kinetic.

### 3.3. 3D Head Pose Estimation

The head pose estimation block A) from Figure 2 combines visual odometry and a revised version of our previous MARG-filter framework. The visual odometry estimation is based on a visual SLAM algorithm (ORB SLAM 2) [21]. The MARG-sensor orientation estimation filter framework is explained in detail in [12,24].

The proposed data fusion addresses the environmental challenges which arise from the teleoperation task and is not reliably solvable using only a single sensing technology. For example, the orientation estimation from MARG-sensors is based on the numerical integration of angular rate measured through the gyroscope. These low-cost consumer grade MEMS based gyroscopes suffer from DC-offsets, known as gyroscope bias. This bias leads to a drift in the integrated angles. This drift is typically compensated by using global references, i.e., direction of gravity and geomagnetic field of the earth measured by the accelerometer and magnetometer, in the data fusion process. However, the measurements are subject to external disturbances effecting the measured direction of the reference vectors and therefore leading to orientation estimation errors. This is especially the case for the measurement of the geomagnetic field vector used to correct heading estimation errors. Indoor scenarios and the presence of ferromagnetic and permanent magnetic materials (e.g., robotic systems) will lead to varying magnetic field vectors which degrade the effect from the geomagnetic vector measurement on the heading correction. Within this work we use visual odometry data in the data fusion process of the MARG-sensor to apply heading correction and improve absolute orientation estimation in indoor scenarios. On the other hand, the sole use of visual odometry is not robust related to the proposed scenario. Using vision based techniques only (e.g., optical flow) would also result in accumulations of errors since the visual scenery will be exposed to a lot of relative motion from the robotic system. Robust visual odometry is based on the dominance of static feature points over moving objects and therefore degrades in the presence of moving objects, in this case the moving robot system.

To address these issues this work utilizes a visual SLAM approach to (a) create an accurate map of the working area to relocalize within the map based on the detected and matched features in order to increase accuracy and robustness and (b) fuse the measurements with MARG-sensor data to reduce the impact of relative motion in front of the cameras visual scene on the orientation estimation and (c) to be able to reset to a known orientation based on the discrete MARG-sensor estimations. The head pose estimation block fuses visual and inertial sensor readings to form a robust pose estimation of a users head without the need for external marker placement, i.e., fiducial markers. Due to the recent technology and research efforts in camera technologies, depth camera sensors have become small, fast, reliable and affordable when it comes to everyday use. Using a depth camera over a regular monocular 2D image sensor adds a complete new dimension and has major advantages when it comes to pose estimation in general. We therefore use a stereo depth camera as the input measurement for a visual SLAM approach and combine the orientation measurement with a MARG-sensors orientation estimation to generate reliable and robust orientation even under complete loss of visual information. The visual position estimation is used as true head position since MARG-sensors are known for accumulating errors upon estimating translation from double integration of acceleration and might therefore lead to wrong position estimation.

#### 3.3.1. Visual SLAM Based Pose Estimation

The visual SLAM framework is part of block A, the head pose estimation pipeline in Figure 2. In this work we use a state of the art and open source V-SLAM approach, namely ORB-SLAM 2 [21]. This algorithm has proven to be very robust and precise regarding position and orientation estimation, which is in general referred to as pose. The V-SLAM approach uses ORB (Oriented FAST and Rotated BRIEF) to detect and track features from an RGB-D input image.

We use the RGB-D based implementation of ORB-SLAM 2 but instead of supplying the RGB image, we use the infrared image of the first infrared camera sensor from the stereo infrared module of the RealSense™ d435 camera (Intel Corporation, Santa Clara, CA, USA). This image is the basis for the depth image alignment. This avoids the need for an alignment step between RGB and depth image. Furthermore, the RGB camera is rolling shutter whereas the infrared cameras have global shutter and will therefore contain less motion blur. The infrared cameras also features a wider field of view (FOV) compared to the RGB camera (H × V × D-Infrared: $91.2^{\circ }\times 65.5^{\circ }\times 100.5^{\circ }$, vs. RGB: $64^{\circ }\times 41^{\circ }\times 72^{\circ }$ [23]). Using the infrared image decreases the data package throughput send out by the camera. The d435 camera provides the possibility to toggle the laser emitter projector between two consecutive frames. Thus, one image frame is taken with and the next frame without the emitter projector. Images without emitter projector are used as the 2D-image input source to the ORB-SLAM framework, whereas the depth images are provided by the depth estimations from the images with emitter projection. This is enabled by calculating the mean image brightness and selecting the image with lesser brightness as the 2D-image source and the one with higher brightness as the depth source for an increased depth image density, respectively. Additionally, this dense depth image is used as the input to the gaze mapper, compare Figure 2.

Figure 3 shows the image pipeline outputs with and without emitter projector for infrared and depth streams. Using the above described image pipeline decreases the necessary data package size that needs to be handled by the host computer. Furthermore, this procedure ensures a wider FOV image for the visual SLAM algorithm and gaze mapping instead of using the RGB image. The visual SLAM framework will locate and track ORB features in the current infrared and associated depth image and inserts keyframes into a new map. Based on epipolar geometry and fundamental matrix construction the camera pose (orientation and position) is estimated using a constant velocity model between consecutive frames and optimized through bundle adjustment. ORB SLAM is capable of loop closure during map creation and furthermore relocalizes the camera when re-visiting known locations inside the generated map [21]. The ORB-SLAM framework features a localization only mode to reduce computational costs that can be enabled if a sufficient map has been captured.

This algorithm is capable of generating reliable position and orientation data while visual frames are available. If a sufficient large map has been created (>50 keyframes) we enable the localization only mode to reduce computational costs and reduce pose estimation errors from relative motion in the visual scenery. To further enhance robustness, we disable the visual odometry constant velocity motion model within the localization mode. This ensures that the pose estimation does rely on matched feature points in the map only and does not interpolate between unmatched features using the velocity model. On the one hand this procedure ensures that the visual pose estimation is less error-prone to relative motion in the scene. On the other hand the overall tracking robustness of ORB-SLAM is reduced which will result in localization failures during strong dynamic motion. If ORB-SLAM fails to localize in the scene the mapping mode is enabled again. The visual SLAM framework passes the visual orientation data to the MARG-filter for data fusion purposes described in Section 3.3.2. The visual SLAM framework directly provides the head position data to the complete head pose estimate, see Figure 2. There is no need to fuse the position data from the visual SLAM framework, as the position estimation from the MARG-sensor is not reliable due to drift. The manufacturing immanent DC-offset of the MEMS accelerometer leads to a second order drift phenomenon based on the double integration of accelerometer raw sensor data. The orientation estimation from the visual SLAM approach however is passed to the MARG-filter framework. A reliable orientation calculation is provided by the MARG-filter framework even when visual feedback is lost or compromised by relative motion, e.g., the robotic system moving through the scenery. The MARG-filter framework bridges the downtime of the visual data and furthermore reinitializes the visual SLAM algorithm by passing the current orientation as the initial state.

#### 3.3.2. Visual-Inertial Orientation Fusion

The second algorithm to robust pose estimation utilizes MARG-sensor measurements fused with visual heading information from the V-SLAM approach. The fusion of data are used to calculate an orientation estimation. Figure 4 gives an overview of the proposed data fusion approach. Synchronization between the MARG and V-SLAM orientation data is achieved based on a ROS node using an approximate time policy matching the different sensors timestamps. The node uses the ROS message_filter package to synchronize the data packets by finding the minimal timestamp difference from a queue of 20 samples (MARG and Camera samples). The best matched samples are used to transform and align the V-SLAM orientation into the MARG-sensor coordinate system. This V-SLAM based aligned orientation is used to calculate the visual heading information ${\vec{N}}_{v,k}$ which is passed to the MARG-filter framework, enabling the calculation of a full quaternion ${}_{B}^{N}q$ representing the users head orientation estimation. Even if visual information is lost for a longer period of time (up to 25 s) or the visual information is degraded because of high dynamic relative motion in the scene, e.g., the moving robotic system, data fusion allows for a robust orientation estimation.

The following section first explains the MARG filter framework and secondly introduces the data fusion between visual heading information and MARG-sensor orientation estimation in more detail. The data fusion process consists out of a dual stage filter that incorporates a gradient descent filter stage (GDA), calculating a correction quaternion based on reference sensors (accelerometer and magnetometer) and fuses it with a gyroscope based prediction quaternion inside a linear Kalman filter (KF). The complete filter derivation is given in [12]. The first filter stage is based on Madgwicks popular method for calculating orientation by solving a minimization problem that rotates a unit vector into a measurement vector
(1)$$f\left({}_{B}^{N}q,{}^{N}\vec{d},{}^{B}\vec{s}\right)={}_{B}^{N}q\cdot \left(\begin{array}{c} 0\\ {}^{N}\vec{d} \end{array}\right)\cdot {}_{B}^{N}\dot{q}-\left(\begin{array}{c} 0\\ {}^{B}\vec{s} \end{array}\right),$$
where ${}_{B}^{N}q$ is a quaternion, ${}^{N}\vec{d}$ unit vector and ${}^{B}\vec{s}$ the measurement vector.

The solution proposed by Madgwick et al. [26] is based on gradient descent algorithm and can be expressed in general form as
(2)$${}_{B}^{N}q_{k+1}={}_{B}^{N}q_{k}-\mu_{t}\frac{\nabla f({}_{B}^{N}q_{k},{}^{N}\vec{d},{}^{B}\vec{s})}{\left||\nabla f({}_{B}^{N}q_{k},{}^{N}\vec{d},{}^{B}\vec{s})|\right|},\;k=0,1,2,\ldots ,n,$$
where $\mu_{t}$ represents the gradient step size.

The MARG-filter framework transfers the iteratively updated quaternion, calculated at the onboard MCU of the MARG-sensor, to the visual SLAM algorithm. The quaternion is split into a roll and pitch quaternion as well as a yaw quaternion. The roll and pitch quaternion is directly used as robust input for orientation information, whereas the yaw quaternion is corrected within the visual SLAM framework using the V-SLAM based heading vector. This is because the yaw quaternion is subject to drift originating from the gyroscope offset, which dynamically changes over time due to temperature and mechanical packaging conditions. Within this work the gyroscope offset drift in heading direction is corrected by applying a set of equations to calculate the visual heading vector within the RGB-D odometry framework. In our previous work we proposed the prior mentioned set of equations to calculate an IMU heading vector and apply it to a GDA based filter while magnetic disturbance is present [24]. The IMU heading vector substitutes the magnetometer vector and therefore reduces the needed sets of equations and guarantees convergence as well as a continuous quaternion solution to the minimization problem. Furthermore, we use an updated form of Madgwick’s GDA equations from [27] that decouples the heading vector update from the pitch and roll update calculation and therefore enhances robustness when the heading vector is disturbed. The process to calculate the visual heading vector substitute is as follows.

The quaternion ${}_{C}^{N}q_{V,k}$ from the V-SLAM algorithm is transformed into the MARG-sensor body orientation through two-sided quaternion multiplication
(3)$${}_{B}^{N}q_{V,k}{=}_{B}^{N}q_{0}\cdot \left({}_{B}^{C}q_{rig}\cdot_{C}^{N}q_{V,k}\cdot_{B}^{C}{\dot{q}}_{rig}\right),$$
where ${}_{B}^{C}q_{rig}$ is the rigid transformation from the origin of the first IR cameras coordinate frame to the MARG-sensor body frame located at the center of the MARG-sensor housing, $\dot{q}$ represents the conjugate quaternion, respectively, and ${}_{B}^{N}q_{0}$ is the initial quaternion that aligns the visual orientation estimation in the navigation frame with the MARG-sensor orientation.

The heading part of the transformed visual quaternion is extracted using the following process
(4)$$\begin{array}{c} q={\left(\begin{array}{cccc} \left(\left({q_{V,k,1}}^{2}+{q_{V,k,2}}^{2}-{q_{V,k,3}}^{2}-{q_{V,k,4}}^{2}\right)\right) & 0 & 0 & \left(2\cdot \left(q_{V,k,2}\cdot q_{V,k,3}+q_{V,k,1}\cdot q_{V,k,4}\right)\right) \end{array}\right)}^{T},\\ {}_{B}^{N}q_{\psi }=\frac{q}{∥q∥}\\ q{=}_{B}^{N}q_{\psi }+{\left(\begin{array}{cccc} 1 & 0 & 0 & 0 \end{array}\right)}^{T},\\ {}_{B}^{N}q_{\psi ,v}=\frac{q}{∥q∥}. \end{array}$$

Secondly a roll and pitch quaternion is calculated based on the iteratively updated orientation estimation from the MARG-sensor by conjugate quaternion multiplication of the heading quaternion from the MARG-sensor and the current output quaternion, to get rid of the heading rotation
(5)$${}_{B}^{N}q_{\phi ,\theta ,k}{=}_{B}^{N}{\dot{q}}_{\psi ,k}\cdot_{B}^{N}q_{k}.$$

A new quaternion is formed that represents the complete visual heading quaternion by quaternion multiplication from the visual heading and the roll and pitch quaternion
(6)$${}_{B}^{N}q_{vh,k}{=}_{B}^{N}q_{\psi ,v}\cdot_{B}^{N}q_{\phi ,\theta ,k}.$$

This quaternion is now used to directly transform an x-axis unit vector into the visual heading vector by quaternion multiplication
(7)$$\begin{array}{c} \vec{x}={\left(\begin{array}{ccc} 1 & 0 & 0 \end{array}\right)}^{T}\\ \left(\begin{array}{c} 0\\ {}^{B}{\vec{N}}_{v,k} \end{array}\right){=}_{N}^{B}q_{vh,k}\cdot \left(\begin{array}{c} 0\\ \vec{x} \end{array}\right)\cdot_{N}^{B}{\dot{q}}_{vh,k} \end{array}.$$

The visual heading vector ${}^{B}{\vec{N}}_{v,k}$ is used as a complete substitute to the magnetometer north heading vector inside the GDA stage forming a complete and continuous quaternion solution. The quaternion from the GDA is now applied as measurement inside the update step of the linear Kalman filter to correct for orientation accumulation errors from gyroscope bias, see Figure 4 KF.

The proposed mechanism of calculating a substitute for the GDA heading vector is not limited to the visual heading vector substitute presented here. In the case of degraded data from the RGB-D odometry framework, e.g., visual occlusion, the procedure enables the use of IMU or magnetometer data for the calculation of the heading vector. Switching in between the three heading vector modes based on visual, magnetic and inertial data, respectively, allows robust heading estimation based on the current availability and reliability of the different sensor measurements. In the following inertial and magnetic heading vector calculation is presented in more detail.

The method for calculating the heading vector ${}^{B}{\vec{N}}_{IMU}$ given by IMU data is similar to the heading vector ${}^{B}{\vec{N}}_{v}$ given by visual data. This is achieved by substituting ${}_{B}^{N}q_{V}$ in Equation (Equation 4) with the Kalman filter output quaternion ${}_{B}^{N}q_{k}$ and calculating the IMU heading vector through Equations (Equation 5)–(Equation 7). The process isolates the heading component for the transformation quaternion in Equation (Equation 6) which allows to sample and hold the current heading orientation if heading rotation is not exceeding a certain motion condition, e.g., angular rate slower than $0.01^{\circ }s^{-1}$.

The magnetic heading is calculated based on cross product between the measured gravity and magnetic field vectors from the MARG-sensor. More details are given in reference [24].

Regardless of whether the heading vector is calculated based on visual, IMU or magnetic data it represents redundant information perpendicular to the plane defined by the pitch and roll component. However, in the case of disturbance of any data source the other sensors are used to calculate the heading vector. The result is a robust and complete orientation estimation under various disturbances.

The filter switches between the heading sources based on vector scalar product of the heading vector based on visual, IMU or magnetic data and the current output of the Kalman filter heading estimate
(8)$$\epsilon_{h}=arccos\left({}^{B}{\vec{N}}_{k}\cdot^{B}{\vec{N}}_{h}\right),$$
where · represents the scalar product and ${}^{B}{\vec{N}}_{h}$ is to be substituted with either visual, magnetic or IMU heading vectors. Based on the relative deviation (quantity of $\epsilon_{h}$) the filter switches towards the appropriate heading vector input
(9)$${}^{B}{\vec{N}}_{k}=\left\{\begin{array}{cc} {\vec{N}}_{v} & \mathrm{if}\;\left(\epsilon_{v}<\epsilon_{m}\wedge \epsilon_{v}<th\right)\\ {\vec{N}}_{m} & \mathrm{if}\;\left(\epsilon_{m}<\epsilon_{v}\wedge \epsilon_{m}<th\right)\\ {\vec{N}}_{IMU} & \mathrm{otherwise}\; \end{array}\right.,$$
where th is a predefined threshold.

The filter selects the most reliable heading source from the relative deviations (ϵ) and availability of the heading sources in the current conditions. The fusion process presented in this work ensures a fast, robust and continuous quaternion solution to be found under various disturbances of data sources. The filter selects the visual heading source under static and slow dynamic motion conditions since it delivers accurate heading information and is capable of correcting drift accumulation. During fast dynamic motion IMU measurements are selected as heading vector information. This is because of the V-SLAM motion estimation artifacts caused by latency issues of the visual SLAM pipeline due to the low sampling frequency of the camera measurements 30 Hz. The filter switches towards either magnetic or IMU heading vector if the visual heading is not available. If magnetic or IMU heading vectors are used depends on the respective deviation angle (ϵ), see reference [24] for details. If the visual heading vector is lost due to the V-SLAM frameworks inability to relocalize in the map within five seconds, the V-SLAM mapping process is resetted. During this time the filter relies on IMU and/or magnetic data from the MARG-sensor. Once the V-SLAM algorithm is restarted the current MARG-sensor orientation is sampled and used to transform and align the orientation estimation into the common navigation frame of the MARG-sensor
(10)$${}_{B}^{N}q_{0}={\;}_{B}^{N}q_{k+1}$$

The visual orientation estimation is transformed into the MARG-sensor coordinate frame based on Equation (Equation 3).

The presented data fusion process is implemented on the custom MARG-sensors MCU running at 300 Hz ensuring low latency between data fusion updates and MARG-sensor measurements, while only incorporating visual feedback into the filter if it meets the before mentioned motion conditions.

#### 3.3.3. Visual Position Estimation

The visual position estimation from the V-SLAM algorithm is transformed into the MARG-sensor navigation frame based on the fused orientation estimation from the visual inertial orientation estimation, see Section 3.3.2.

The translation vector is transformed into the MARG-sensor navigation frame based on the following process
(11)$$\begin{array}{c} \left(\begin{array}{c} 0\\ {}^{B}{\vec{t}}_{k} \end{array}\right){=}_{C}^{B}q_{rig}\cdot \left(\begin{array}{c} 0\\ {}^{C}{\vec{t}}_{k} \end{array}\right)\cdot_{C}^{B}{\dot{q}}_{rig},\\ \left(\begin{array}{c} 0\\ {}^{N}{\vec{t}}_{k} \end{array}\right){=}_{B}^{N}q_{0}\cdot \left(\begin{array}{c} 0\\ {}^{B}{\vec{t}}_{k} \end{array}\right)\cdot_{B}^{N}{\dot{q}}_{0}+\left(\begin{array}{c} 0\\ {}^{N}{\vec{t}}_{0} \end{array}\right), \end{array}$$
where ${}^{N}{\vec{t}}_{0}$ is the last known position estimate that is sampled upon reset of the mapping process
(12)$${}^{N}{\vec{t}}_{0}{=}^{N}{\vec{t}}_{k+1}.$$

Equations (Equation 3) and (Equation 11) describe the full pose transformation from camera to MARG-sensor coordinate frame which is denoted as ${}_{B}^{N}T$ for readability (see Figure 4).

The presented fusion approach allows for robust orientation estimation even if visual feedback is lost or magnetic disturbance is present and therefore enables robust head pose estimation which is key for mobile and accurate gaze based robot control.

The estimated head pose in the MARG-sensor navigation frame is transformed into the robots coordinate system to allow for direct orientation and position estimation in the applications Cartesian space.

### 3.4. Three Dimensional Gaze Point Estimation

The proposed interface is designed to generate two different gaze signals: eye-gaze and head-gaze, respectively. First eye-gaze mapping is described followed by gaze transformation, see Figure 5 upper part. Secondly head-gaze mapping and the respective real world transformation is described, see Figure 5 lower part.

Obtaining accurate eye-gaze data strongly depends on the eye to world camera calibration. Three-dimensional eye-gaze estimation from binocular gaze ray intersection are heavily dependent on the eye model and calibration accuracy [8]. Instead of using a 3D gaze vergence model between both eye tracker cameras, we use a standard 2D calibration based on polynomial mapping to calibrate binocular pupil positions onto a cameras image. The gaze mapper tracks a fiducial marker at five different locations (e.g., on a computer screen or presented by hand) and samples gaze pixel coordinates from the eye cameras alongside world pixel coordinates of the fiducial marker. The parameters of the second degree polynomial are calculated from standard five point target calibration using singular value decomposition for binocular gaze data [28]. The gaze mapper consists of two custom ROS nodes that synchronize the pupil detection results with the RealSense infrared image stream and furthermore enable the AprilTag based eye-gaze calibration routine. This procedure ensures, that the RealSense camera port is not blocked by a single application and can be accessed by all nodes inside the ROS network, i.e., ORB-SLAM node, Pupil service and infrared image synchronization as well as AprilTag detection.

In this work a lightweight binocular eye tracker with an USB-C mount from Pupil Labs is used. We use the pupil labs open source software pupil service [25]. The pupil service client provides the pupil detection pipeline which is then used inside the gaze mapper. The gaze mapper uses the filtered 2D infrared image stream (no emitter, see Figure 3) as the calibration target image. The calibrated 2D gaze pixel coordinates on the 2D infrared image are used to get the gaze vectors magnitude from the aligned 3D stereo depth image. This single real world depth value is transformed into a 3D vector ${}^{B}\vec{d}$ in the camera coordinate frame by using point cloud reconstruction from the 2D pixel coordinates alongside the real world depth value into a 3D point using the cameras intrinsic parameters.

Using the pinhole camera model without lens distortion (see Figure 6a), a 3D point is projected into the image plane using a perspective transformation
(13)$$\left(\begin{array}{c} u\\ vs.\\ 1 \end{array}\right)=\left(\begin{array}{ccc} f_{x} & 0 & c_{x}\\ 0 & f_{y} & c_{y}\\ 0 & 0 & 1 \end{array}\right)\left(\begin{array}{cccc} r_{11} & r_{12} & r_{13} & t_{1}\\ r_{21} & r_{22} & r_{23} & t_{2}\\ r_{31} & r_{32} & r_{33} & t_{3} \end{array}\right)\left(\begin{array}{c} X\\ Y\\ Z\\ 1 \end{array}\right).$$

Leaving out the perspective transformation, assuming the camera coordinate is the origin, we can rewrite it to the following:(14)$$\left(\begin{array}{c} u\\ vs.\\ 1 \end{array}\right)=\left(\begin{array}{ccc} f_{x} & 0 & c_{x}\\ 0 & f_{y} & c_{y}\\ 0 & 0 & 1 \end{array}\right)\left(\begin{array}{c} X\\ Y\\ Z \end{array}\right)$$

Since we get the pixel coordinates u,v in the infrared stream from the mapped gaze point we can directly select the real world depth value *Z* from the depth image stream which is aligned to the infrared image stream. Having u,v and *Z* we can rearrange and reduce the equation to get the *X* and *Y* coordinates
(15)$$\begin{array}{c} X=\frac{(u-c_{x})*Z}{f_{x}},\\ Y=\frac{(v-c_{y})*Z}{f_{y}},\\ {}^{B}\vec{d}={\left(\begin{array}{ccc} X & Y & Z \end{array}\right)}^{T}. \end{array}$$

Head pose estimate and gaze mapper outputs are the input variables for the 3D gaze transformer, see Figure 5.

The 3D vector ${}^{B}\vec{d}$ is transformed from the local camera coordinate system into the world coordinate frame by using the perspective transformation which is the estimated head pose (${}_{B}^{N}q_{k}{,}^{N}{\vec{t}}_{k}$) in the robots coordinate frame from the visual-inertial pose estimation, see Figure 6b. The full head pose transformation from MARG-sensor to robot coordinate frame is given through Equations (Equation 3) and (Equation 11), substituting the static quaternion (${}_{B}^{N}{\dot{q}}_{rig}$) with the one that transforms from the MARG-sensor to the robots’ coordinate frame and set the initial alignment pose (${}_{B}^{N}{\dot{q}}_{0}$,${}^{N}{\vec{t}}_{0}$) in the robots coordinate frame. This transformation can be given either by providing an absolute position estimate in the robots coordinate system, e.g., using fiducial marker detection, or by using a three point initial setup routine that defines the robots coordinate system. To perform initial pose estimation the user needs to focus the laser at three dots to define the x and z axis of the coordinate frame center that are known in the robots coordinate system—e.g., focusing the base of the robot to align coordinate frames. The y-axis is calculated from the cross product between the defined x and z-axis.

Substituting the rotation matrix from the perspective transformation by two-sided quaternion multiplication results in the following formula for 3D world gaze estimation
(16)$$\begin{array}{c} \left(\begin{array}{c} 0\\ {}^{N}\vec{d} \end{array}\right){=}_{B}^{N}q_{k}\cdot \left(\begin{array}{c} 0\\ {}^{B}\vec{d} \end{array}\right)\cdot_{B}^{N}{\dot{q}}_{k}+\left(\begin{array}{c} 0\\ {}^{N}\vec{t} \end{array}\right),\\ {}^{N}\vec{d}={\left(\begin{array}{ccc} x & y & z \end{array}\right)}^{T}. \end{array}$$

Using the above mentioned setup allows for accurate 3D eye-gaze estimation in a working area, that is restricted by the RGB-D cameras depth resolution. The gaze mapping method will only correspond to the pixel value chosen.

The interface can also be used for head-gaze based 3D point estimation. In this mode a small eye safe laser mounted above the camera is used for direct user feedback. The laser indicates the head-gaze vector endpoint that is used for vector magnitude estimation and world point transformation. The laser pointers pixel position in the camera frame is calculated based on Equations (Equation 15) and (Equation 16). Since the lasers coordinate system and camera coordinate system are not perfectly aligned, the physical displacements and rotation from the laser pointer with respect to the camera center and projection of the depth value on the surface need to be incorporated. The depth value *Z* from vector ${}^{B}\vec{d}$ is calculated based on the known orientation of the camera with respect to the world coordinate frame and is projected using trigonometric relations of the predicted real world coordinates of the laser dot with respect to the camera frame (*B*). This results in the following set of equations to calculate the projected depth $Z_{l}$ from the center pixel (px,py) and the *x* and y-axis displacement of the laser pointer,
(17)$$Z_{l}=Z\left(px,py\right)+\Delta Z_{x}+\Delta Z_{y}$$
where $\Delta Z_{x}$ and $\Delta Z_{y}$ are calculated as
(18)$$\begin{array}{c} \Delta Z_{x}=\frac{X_{l}}{tan(\pi -\theta )}\\ \Delta Z_{y}=\frac{Y_{l}}{tan(\pi -\psi )}, \end{array}$$$X_{l}$ and $Y_{l}$ are the physical displacements between the laser pointers’ center and the cameras center, while θ,ψ are the pitch and yaw Euler angles acquired from the cameras’ orientation in the world frame (*N*), respectively. The laser pointers’ pixel position is calculated based on Equation (Equation 15) where the input vector ${}^{B}{\vec{d}}_{l}$ is
(19)$${}^{B}{\vec{d}}_{l}={\left(\begin{array}{ccc} X_{l} & Y_{l} & Z_{l} \end{array}\right)}^{T}.$$

### 3.5. Robot Control

There are various strategies for control of a robot in 3D space by head motions. In this work we use two control strategies to precisely control the robots EFF in Cartesian space in all 6 DoF’s. The first control strategy uses head- or eye-gaze to control the robots EFF’s position in 3D Cartesian space, while the orientation stays fixed. The second mode utilizes head motion to control the robots EFF’s orientation. This strategy employs a motion mapping between the 3 DoF of the humans head rotation onto the robots EFF orientation. This is based on the work from Rudigkeit et al. [1,6]. The user can switch towards the appropriate control mode by a simple eye blink which toggles the state variable $S_{t}$, see Figure 7. The control strategies are depicted in Figure 7 and are explained in detail below.

In position control the 3D gaze point (${}^{N}\vec{d}$) is fed to a PID (Proportional Integral Derivative) controller. The error term is calculated from the desired 3D point (head- or eye-gaze) and the robots EFF position, which is calculated based the inverse kinematics (IK) equation. The output is a Cartesian velocity vector which is saturated to enforce speed limitations onto the robot for increased safety. This vector is feed to a velocity based Cartesian jogger. The jogger calculates the desired joint positions and velocities and publishes both to the robots ROS interface which communicates with the robots internal controller. The robots joint angles are used to calculate the EFF’s new pose through its forward kinematics (FK). The new pose is looped back to calculate the new error term. This simple PID jogging allows for smooth and continuous 3D position robot control in Cartesian space. During jogging, the robots EFF orientation is held constant since orientation is not represented by the 3D head- or eye-gaze point.

In orientation control the pitch, roll and yaw angles given by the users’ head motion is mapped to the 3D orientation of the robots EFF. This motion mapping has been intensively studied and allows for precise control of a robots EFF orientation [1,6]. Switching from position to orientation control and vice versa is based on blinking with the right eye. Upon changing from position to orientation control, the current orientation is sampled and used as zero orientation baseline with a $15^{\circ }$ deadzone (see Figure 7). The user needs to rotate the head to the desired angle (pitch, roll, yaw) beyond the deadzone threshold to rotate the EFF. The angular velocity of the EFF’s orientation change scales with respect to the relative angle change. A bigger angle equals a higher angular velocity whereas a small angle results in low angular velocity. By blinking again, the motion state is switched back to position control. This setup allows for full continuous Cartesian motion control for position and orientation of the robotic system.

## 4. Experimental Setup

The experimental setup is designed to evaluate (a) the long term heading drift reduction through the proposed visual and inertial data fusion, in contrast to inertial or visual data only orientation estimation, and (b) the short term orientation estimation stability if visual data are not available and (c) proof of functionality of the interface for real world gaze based robot control.

The accuracy of the pose estimation with the proposed interface is benchmarked against an infrared based marker system from Qualisys [30]. Therefore, the user is wearing the interface alongside a rigid marker tree on top of the 3D printed custom case, see Figure 8. The user is sitting in front of a table with a surface area of 0.8 m by 1.2 m. The tables surface is covered with a target marker grid representing the head- and eye-gaze targets. A total of 24 targets is placed on the surface, spaced evenly in a 0.2 m by 0.2 m grid, see Figure 8. The target positions are known with respect to the world coordinate system from the Qualisys system. The current pose of the rigid marker tree is used as the initial pose of the interface and is passed to the head pose estimation pipeline to align the interface pose with the Qualisys pose. To examine robustness of the data fusion approach in magnetically disturbed environments, the magnetometer data and therefore magnetic heading correction is turned off.

For eye-gaze accuracy tests a single calibration marker (AprilTag [31]) is presented to the user at five different locations to map the pupil positions onto the infrared cameras stream as described in Section 3.4. After calibration, which takes about 30 s, the user randomly focuses different targets with a fixation time of around 1 s and without restrictions regarding head motion. The user presses and holds a mouse button upon fixating a target marker to trigger a record function that indicates the eye or head-gaze is on the target. Upon button release, the trigger signal is turned off which stops the record function for this specific gaze target. The mouse button press and release ensures, that the users intended eye or head-gaze is on the target markers and thus rejects artificially introduced errors between target motions. The user is encouraged to move around with the chair in front of the table to evaluate robustness of the presented interface under dynamic motion conditions. A single trial takes 20 min in total without the calibration process.

The same setup is used to investigate head-gaze accuracy. The laser diode is turned on to give the user feedback of the current head-gaze point. Likewise, to the eye-gaze experiment the user starts focusing targets with the laser dot for around 1 s without restrictions regarding head motion or position and toggles the mouse button when the target is in focus.

The data streams of the proposed interface (orientation, position, head/eye-gaze point and on target event) and the ground truth motion capture data (orientation, position, target locations) are synchronized via timestamp based filtering from a custom ROS-node. The recording node ensures that maximum latency between the data streams is about 3 ms in total between the data streams. The data streams are synchronized at a 100 Hz rate.

The proof of concept for the full gaze based control pipeline is tested on a real robotic system. The user randomly gazes at five different target points inside a robots working area for 20 min in total. The user blinks with the left eye to send the gaze point to the robot control pipeline upon which the robot moves to this point. The user sits in front of a dual arm robotic system and is in control of one arm, see Section 5.4 for more details. The target positions are placed at three different heights (14 cm, 2 cm and 0 cm with respect to the table surface) to demonstrate three-dimensional position control. The target positions are known with respect to the robot coordinate system and are compared to the robots tool center point (TCP) position.

## 5. Results and Discussion

The presented data fusion approach from the combination of the MARG-filter and RGB-D SLAM framework’s is evaluated based on the accuracy of the orientation and position estimation and presented in Section 5.1 and Section 5.2. Therefore the orientation and position estimation is benchmarked against an optical marker based reference system [30].

Furthermore this work presents an eye-gaze or head-gaze based modality to move the robot to a desired point in three-dimensional space. Accuracy of the 3D gaze point estimation is also benchmarked against the Qualisys system in Section 5.3. A total of 30 trials was recorded to estimate orientation, position and gaze point accuracy. Orientation and position estimation accuracy is calculated based on the head motion from all 30 trials.

Head- and eye-gaze accuracy are calculated from 15 trials for each gaze vector method, respectively. The data was accumulated for three male participants in the age of 29–35. Two individuals wear glasses.

Lastly the results for gaze based robot control in a real world scenario with a robotic system are reported in Section 5.4. A total of 5 trials was recorded to proof robot control functionality and robustness to relative motion in the scenario.

### 5.1. Visual-Inertial Orientation Estimation Accuracy

The orientation estimation accuracy is calculated as the mean of the RMSE values (root mean squared error) between ground truth Qualisys data, the visual-inertial and inertial only orientation estimations for all 30 trials. RMSE values for visual only data is not calculated since the RMSE will change with respect to the number of visual data losses and the magnitude of error compared to ground truth during these losses. Furthermore, the ORB SLAM framework is resetted if relocalization fails within a time frame of five seconds and will be aligned with the MARG-sensors orientation estimation, compare Section 3.3.2.

The results are presented as Euler angles in degrees, compare Table 1. Throughout all trials the visual-inertial yaw orientation estimation results in a mean RMSE of $0.81^{\circ }\pm 0.44$ after 20 min of continuous head motion. The inertial only orientation estimation on the other hand results in a mean RMSE of $12.49^{\circ }\pm 8.48$ for all trials recorded.

Figure 9a presents typical yaw angle results for the visual-inertial, inertial only, visual only and ground truth orientation estimation for a typical trial. The user randomly gazes at the 24 targets without restrictions to head motion. Figure 9b depicts the absolute error values from the subset. The maximum error for inertial only orientation estimation of this subset results in $35^{\circ }$ accumulated drift after 20 min whereas the visual-inertial orientation estimation results in a maximum error of $3.7^{\circ }$ (compare minute 19.5) and a total drift of $0.5^{\circ }$ compared to the ground truth. While no visual data is available, the visual only yaw angle estimation experiences errors scaling with respect to the absolute orientation change, see, e.g., minute 1,3,9.5 and so on. Visual data loss originates from two different sources. Short period peaks of visual data loss are due to localization failure in the map. This occurs during dynamic motion in the visual scene between consecutive frames and the inability of the SLAM algorithm to relocalize with the current features in the given map. Longer visual data loss plateaus are due to intentional covering of the camera with the hand to prove the heading vector switching mechanism and robustness of orientation estimation during long visual occlusions.

Figure 10 depicts the heading vector switching mechanism during complete loss of visual data for a 6 s and 10 s period from an enlarged segment of the trial from Figure 9a.

The proposed visual inertial orientation estimation pipeline reduces the gyro based drift accumulation for the yaw angle estimate to a minimum compared to the inertial only orientation estimation. During all trials the proposed fusion approach maintains the orientation and results in a mean RMSE drift of $0.81^{\circ }$ in total whereas the inertial only heading estimate results in significant mean RMSE ($12.47^{\circ }$) due to missing heading correction (see Table 1). Pitch and roll angles are calculated based on accelerometer and gyroscope data and are typically less error prone, since these angles are calculated based on the direction of gravity which at least can be measured in slow or static phases to correct for drift.

The proposed visual inertial orientation estimation pipeline significantly reduces accumulation of drift in the heading estimate in magnetically disturbed environments and enables full quaternion based orientation calculation, even if the visual heading vector is not available, compare Figure 10. During loss of visual data the algorithm is capable of switching towards the IMU heading vector substitute to keep the orientation estimation stable. Even during long visual data losses of 40 s (see Figure 9, minute 18.75) the proposed orientation estimation is able to produce reliable orientation data, even though it accumulates drift due to the gyroscope bias. As soon as visual orientation estimation is available again, the filter switches back to incorporate the visual heading vector into the orientation estimation pipeline. If relocalization in the map is successful, the accumulated gyro bias error can be subtracted, which is the case throughout the depicted trial. The proposed fusion scheme is able to switch to IMU only orientation estimation mode and maintain reliable heading estimation until visual feedback is available again. If magnetometer data is available and plausible (small relative deviation), the filter switches towards magnetic heading correction and reduce error accumulation even further. Since the experiment is designed to evaluate the filters’ robustness while magnetic disturbance is present, the magnetic heading correction does not partake in the orientation estimation results depicted here.

### 5.2. Visual Position Estimation Accuracy

Trajectory estimation accuracy is calculated as the mean of the RMSE values of the visual position estimation and the ground truth Qualisys data and is presented in meters. The mean Euclidean position estimation error for the total of 30 trials is 28.0±28.5 mm, compare Table 2. Figure 11a depicts a 3D representation of one subset of head trajectory measurements for the ground truth trajectory (blue) and the estimated trajectory of the ORB-SLAM visual position estimation (orange). In this trial the user moves the head covering a total volume of 0.75 m × 0.8 m × 0.2 m (*x*, *y*, *z*) in total.

Figure 11b depicts the absolute error for the subset in three individual axes, respectively. The maximum absolute error for an individual axis in this trial is 130 mm in the *z*-axis for a short duration of 0.2 s. The total RMSE for the depicted set is 9.1 mm.

The trajectory estimation relies on visual position estimation from the ORB SLAM framework and transformation into the correct coordinate frame based on the visual and inertial based orientation estimation. This setup allows for accurate position estimation while visual pose estimation is available. If visual information vanishes, the last known position is held until visual position estimation is available again. If the local map from ORB-SLAM is sufficient, a relocalization and recovery is possible which will result in a small error in the position estimation. Relocalization is effective for example at minute 6.30 of the trial presented in Figure 11b. The relocalization reduces the accumulated position error of the largest outlier from 0.13 m to 0.01 m in the z-Axis. If the track is lost however, the map is resetted which might introduce a position error that depends on the length of the visual feedback outage. The maximum length of visual outage without relocalization is limited to 3 s. Since the interface is worn by a human during robot teleoperation, the overall position change during a possible visual outage is rather limited and hence does not lead to large position errors. Furthermore, a map for a typical human robot shared workspace is typically small which allows for fast mapping and small maps, which in turn helps with relocalization.

### 5.3. Three Dimensional Gaze Point Estimation Accuracy

Gaze point accuracy is divided into two subsections, head- and eye-gaze accuracy, respectively. The accuracy for either method is calculated based on mean error values between ground truth values of the target points from Qualisys measurements and the estimated head or eye-gaze point on the target. A mean gaze point is calculated for each of the 24 targets. A total of 30 trials was recorded, 15 for either head and eye-gaze target positions.

#### 5.3.1. Head-Gaze

Figure 12a depicts a typical subset to illustrate head-gaze point accuracy for the x and y plane. The ground truth position for each target is presented as red circles whereas the head-gaze points from this subset are depicted as blue circles. Every gaze target is focused multiple times during the trial, hence there are multiple gaze points (blue circles) for each target point. The head-gaze trajectory for a whole target transition cycle between all 24 targets is depicted as a dotted black line. Several gaze points are distributed around the ground truth target point. The maximum head-gaze position error for this trial is 30.0±20.0 mm in the x-Axis, 18.0±12.0 mm in the y-axis and 12.5±8.0 mm in the z-axis. Figure 12b depicts mean head-gaze position error and its standard deviation for all 24 targets throughout all 15 trials. The mean Euclidean head-gaze error for all trials results in of 19.0±15.7 mm, see Table 3.

The head-gaze error is increasing with distance between the participant and the target. Target points at the more distant end of the table have a higher standard deviation and larger mean error compared to the targets in the front of the table (see Figure 12b). This is due to the scaling impact of orientation transformation errors, which have a high impact on the gaze point estimation at large distances with respect to the surface. The magnitude of error scales with the pitch angle relative to the surface and therefore a deviation of $0.5^{\circ }$ at 0.3 m height results in a 1 mm error at a 0.1 m distance in the x-axis but results in 190 mm error at 1.0 m distance in the x-axis. The head gaze accuracy is furthermore reduced due to human errors when aiming for the targets. If the head-gaze point (feedback laser) is not exactly coincident with the target midpoint, artificial errors are introduced that enlarge the standard deviation of the overall accuracy. Nevertheless, the head-gaze accuracy is relatively high, see Table 3.

#### 5.3.2. Eye-Gaze

Besides head-gaze the interface and data fusion process presented here also enable three-dimensional eye-gaze position estimation.

Figure 13a depicts a typical subset for eye-gaze position estimation. The ground truth position for each target is presented as red circles whereas the mean target for the trial is depicted as blue circles. Every gaze target is focused multiple times during the trial, hence there are multiple gaze points (blue circles) for each target point. The eye-gaze for one target transition cycle is depicted as a dotted black line. The largest eye-gaze position error for the depicted trial is 50 mm in the x-axis for target point 23. This target is in the last row of the table-top with the greatest distance to the user. The mean distance between these targets and the user is 1.1 m. The maximum y-axis error for this trial is 41 mm at target point 1. Likewise to the head-gaze experiments, artificial errors are introduced from the user if the gaze point is not coincident on the target point, which in turn enlarge the standard deviation of the overall accuracy. Figure 13b depicts mean eye-gaze position errors and its standard deviation for all 24 targets throughout all trials. The mean Euclidean eye-gaze error for all trials results in 27.4±21.8 mm, compare Table 3.

Eye-gaze point estimation is less accurate when compared to head-gaze estimation, see Table 3. This is mainly due to eye-gaze calibration inaccuracies which results in an offset or inconsistent map of the actual and calibrated gaze point. These inaccuracies in the gaze point estimation leads to the selection of a wrong depth pixel value which in turn results in a different 3D point in Cartesian space upon transformation. This effect does furthermore scale with respect to the distance between the cameras center and the target, compare Section 5.3.1. The eye-gaze error is increasing with distance from the participant and peaks in 20.0±20.0 mm maximum single axis error for points in the last row (1.1 m from head). Accurate gaze calibration is a prerequisite for 3D gaze point estimation. The eye gaze accuracy does also decrease due to slippage of the headset over time. The different calibration accuracies throughout the trials, slippage and human errors from target aiming will accumulate and result in higher standard deviations compared to head-gaze accuracy, see Table 3. Human control of a robot EFF in a small workspace is enabled by eye-gaze as well as head-gaze point estimation. Head-gaze control could be preferred in a larger workspace and for a more precise control approach. Comparing the presented interface to a recent 3D eye-gaze interface proposed in [2] demonstrates the presented interfaces higher accuracy in terms of total euclidean error (27.4 mm vs. 46.8 mm) but has a lower repeatability (21.8 mm vs. 1.4 mm). This effect is mainly due to the prior mentioned inaccuracies from gaze calibration, slippage and furthermore position tracking accuracy differences. Shaftie et al. use an infrared based motion capture system which gives significantly higher resolution compared to the V-SLAM based position estimation. This can also be seen in Table 2, since we use an infrared motion capture system as ground truth to benchmark the visual position estimation.

### 5.4. Robot Control

This subsection presents a proof of functionality of the head-gaze based control pipeline in a real world robot control application. Figure 14b depicts the workspace and the gaze targets for the robot control. The robotic system consists of a dual arm UR-5 mounted at an angle of $45^{\circ }$ onto a t-beam. Both arms are equipped with a Robotiq 2F-85 gripper. Furthermore, a tabletop is welded to the t-beam, that represents the robots’ workspace. The robot control application provides a complete implementation to control each arm separately or even control both arms simultaneously. For this proof of functionality however, the user only controls a single arm (left side). The motion parameters of the robotic system are limited to ensure a human safe teleoperation and are listed in Table 4. The experiment does not involve any direct human machine contact. This is enforced through the physical distance between the user and the robotic arm which is larger than the maximum stretch limit of the robotic arm in the humans’ direction (1.1 m including the gripper). The user aims at five different waypoints on the tabletop in the workspace. Upon a discrete event (blinking with the left eye) the Cartesian gaze point is transferred to the robot control pipeline from Figure 7, actuating the robotic arm.

Figure 14a depicts a top view reprojection of one trial for ground truth target positions (red dots), the commanded head-gaze points (blue dots) and the robots 3D tool center point trajectory (black line) on the robotic workspace. The mean RMSE head gaze values and the standard deviation for the five target points from a total of five trials are plotted in Figure 15.

The mean Euclidean RMSE for head-gaze based position control results in 26.5±20.9 mm for all five target points inside the test workspace. The total mean RMSE for all five targets is only 7 mm larger compared to the 24 targets head-gaze total mean RMSE that only had very little relative motion in the visual scene. This demonstrates the overall usability of the proposed interface and methods for gaze based robot control under the condition of relative motion of the robotic arm in the visual scene. A link to a video demonstrating the interfaces capability for robotic control with head-gaze and head motion as well as eye-gaze position control is given in the Appendix A.

## 6. Conclusions

This work presents a mobile head-worn interface that enables a user to accurately control a robotic arm in 3D Cartesian space via head or eye-gaze. Furthermore, it enables to control the orientation of the end effector of a robotic arm by using accurate 3D head motion angles. The exclusive use of a camera with V-SLAM method or the exclusive use of MARG-sensors for robust estimation of the heading angle shows different intrinsic features in the time domain. While purely visual data from the V-SLAM method have the disadvantage at higher dynamics, i.e., at strong relative movement of the robot arm in the scene, the IMU-sensors show a drift of the heading angle at very slow movements of the head. This work estimates a robust heading angle by fusion of data from a V-SLAM camera with a MARG-sensor. Even though other interfaces exist, these might be obstructive or to heavy for people with severe physical disabilities, e.g., people suffering from paraplegia. The proposed interface is lightweight and mobile and can be used without the need for external reference markers or static cameras.

The data fusion process generates robust and accurate orientation and position estimation of a users head with respect to a dedicated workspace in indoor environments and is capable of switching between visual-inertial, inertial only and inertial-magnetic orientation estimation, based on reliability of sensor data. The presented data fusion is infrastructureless and therefore not dependent on any external references, e.g., fiducial markers, stationary camera equipment and so on. The data fusion process is capable of delivering robust orientation and position even while subject to significant relative motion from the robotic system.

The head or eye-gaze control enables an intuitive communication channel for robot collaboration that facilitates natural gaze based task interaction. Depending on the desired accuracy and size of the workspace, head or eye-gaze could be used interchangeable. For example, the 3D eye-gaze point could be used to quickly determine a ROI for the robot and switch towards head-gaze for accurate control of the end effector position. If the eye-gaze experiences an offset due to slippage from the eye-tracker framer a user can switch towards head-gaze mode and thus maintain control and safety. To further enhance robot end effector positioning precision a mixed control strategy could be used. The head or eye-gaze could be used to determine an ROI for the robots TCP and afterwards switch to use head motion mapping for precise end effector position. This can be achieved by simply using the head motion mapping approach bypassing the PID controller into the jogger node to move the arm in Cartesian position space. The presented interface allows for a multitude of new interaction strategies which will be elaborated in future work.

## 7. Future Work

The interface enables a variety of new control and teleoperation mechanisms for human robot collaboration. For example, the eye-gaze position estimation and head motion measurement could be used in direct combination to translate and rotate the robots EFF at the same time without switching motion states, i.e., fixating a target object with the eyes and directly rotating the head in the desired orientation.

Furthermore, future research will focus on utilizing the Cartesian gaze point estimation for semi autonomous object grasping. The three-dimensional gaze position output could be used to set a Cartesian region of interest (ROI) for a robotic arm that is equipped with a wrist camera. The estimation of the ROI of the object enables the wrist camera on the robotic arm to accurately localize and grab the object. This procedure could lead to a more natural way of human robot collaboration by just gazing at an object of interest therefore effectively reducing the number of control inputs or motion groups needed.

The proposed interface could also be used in various human robot applications. For example, in robotic painting. The presented interface could be used directly on top of the canvas at any location inside the robots workspace and without the need for a dedicated computer screen. This could represent a more natural way of drawing since the head- or eye-gaze point is directly reflected by the robots TCP motion on the same plane. A further example is assembly of small parts. We are planning on using the interface in an assembly task that utilizes the humans vision and decision-making skills to control a robotic arm in order to separate and grab parts from bulk containers. After separation, the parts are placed inside a magazine that is used by a second robotic arm to autonomously assemble the full assembly-module.

## Figures and Tables

**Figure 1 sensors-21-01798-f001:**
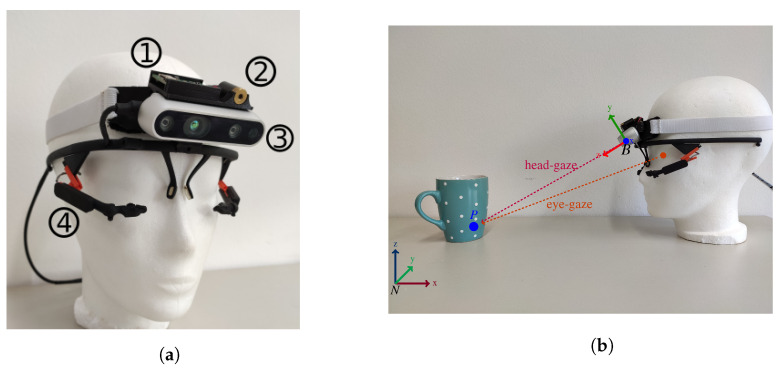
Proposed head interface (**a**) and depiction of head- and eye-gaze vectors (**b**). The head
interface (**a**) consists of a pupil core binocular USB-C mount headset ④ and the custom camera mount
uniting the depth camera ③, MARG-sensor ① and feedback laser ②. Image (**b**) depicts head- and
eye-gaze vector origins from the interface to a world point P.

**Figure 2 sensors-21-01798-f002:**
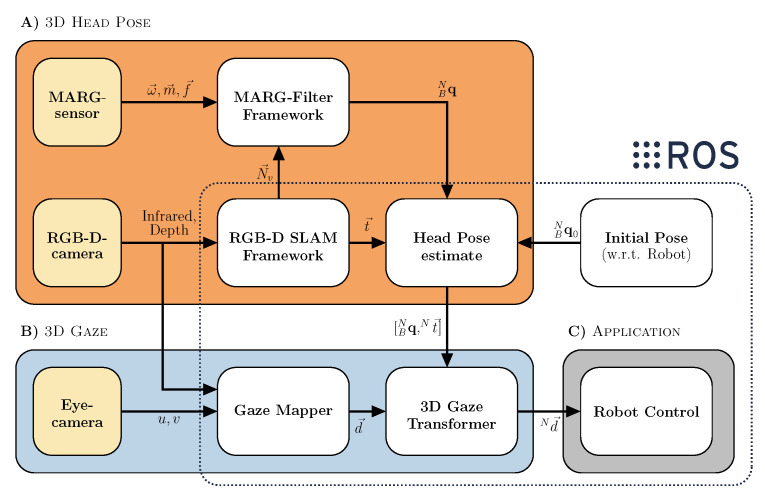
Block diagram of the overall system setup. The system consists of three main Soft- and Hardware components that enable the calculation of robust control signals for robotic teleoperation: (**A**) The accurate head pose estimation based on visual-inertial position and orientation estimation, (**B**) the calculation of 3D eye- and head-gaze from known head pose and gaze points from dense 3D depth images as well as (**C**) the application interface for robot control in 6D Cartesian space.

**Figure 3 sensors-21-01798-f003:**
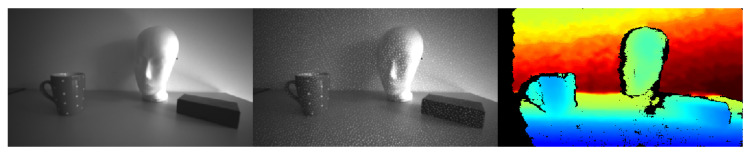
Image sequence as seen by the filtered camera outputs. **Left**, image without emitter projector, **middle**, image with emitter projector pattern (white spots) and **right**, depth image.

**Figure 4 sensors-21-01798-f004:**
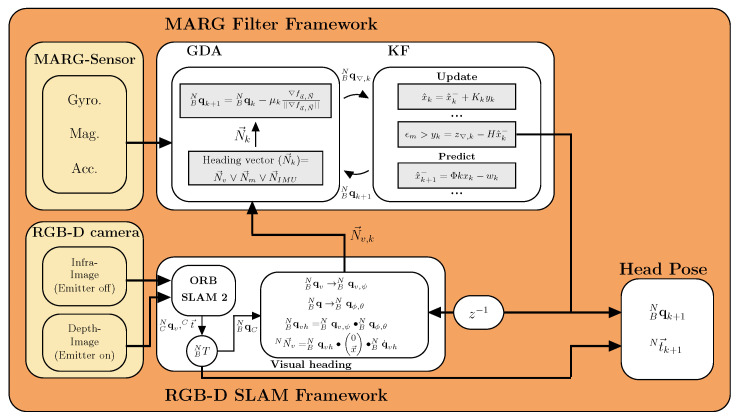
Block diagram of the data fusion software for head pose estimation. The ORB SLAM 2 node calculates a complete pose in the camera frame which is transformed into the MARG-sensors navigation frame. The translation vector ${}^{C}\vec{t}$ is transformed into the world coordinate system, forming ${}^{N}\vec{t}$ and is directly used as the true head position. The head orientation estimation ${}_{B}^{N}q$ is based on the MARG-filter framework incorporating three different correction heading vectors (visual heading, IMU heading, magnetic heading). The filter chooses the appropriate heading source based on disturbance recognition from vector scalar product deviations for robust orientation estimation.

**Figure 5 sensors-21-01798-f005:**
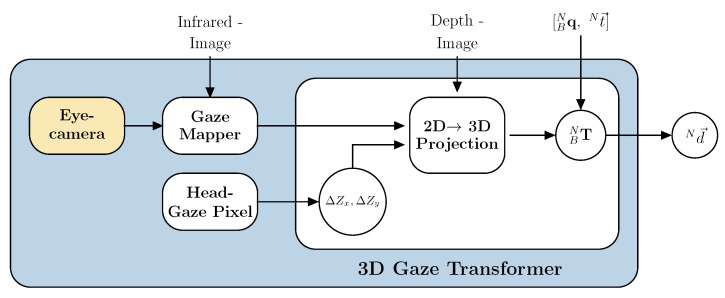
Block diagram of the 3D gaze estimation. The eye cameras measurements are mapped onto the infrared stream to generate a pixel pair whereas the head pose pixel pair is a fixed value. Both pixel pairs are passed to the gaze transformer to reproject the 2D pixel to 3D local camera coordinates. This local vector is lastly transformed into the world coordinate system forming ${}^{N}\vec{d}$.

**Figure 6 sensors-21-01798-f006:**
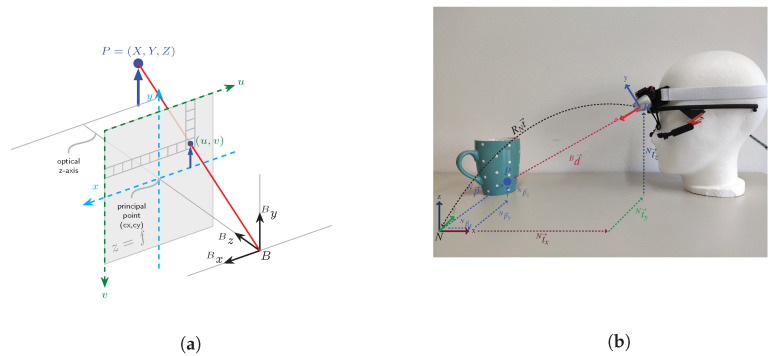
3D gaze vector reconstruction: (**a**) pinhole camera model, adapted from [29] and (**b**)
illustration of gaze point (*P*) depth vector ${}^{B}\vec{d}$ coordinate transformation from body (*B*) to world
coordinate frame (*N*), where *N* is either coincident with the robots origin or the transformation from
*N* to the robots’ origin is known and incorporated into the inverse kinematic chain. The vector ${}^{N}\vec{p}$ is
the input target point for the inverse kinematic calculation of the robotic system.

**Figure 7 sensors-21-01798-f007:**
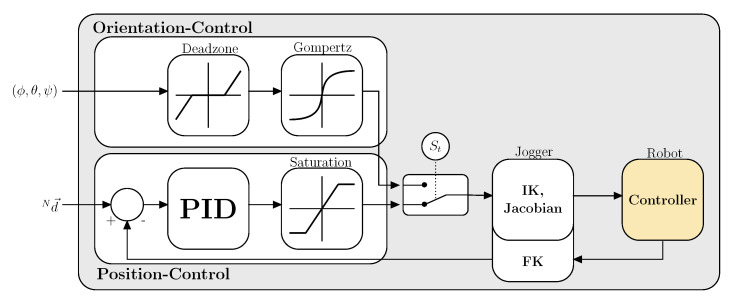
6-DoF Robot control block diagram. The trigger signal $S_{t}$ toggles Cartesian orientation and position control. The toggle is set when an eye blink or other discrete event occurs. The Cartesian position control is generated through a closed loop PID controller with anti-windup and saturation. The error term is calculated from the desired 3D head- or eye-gaze (setpoint) and the robots EFF position from its respective forward kinematics (FK). The orientation of the EFF is mapped based on a gompertz function if the head angles exceed a certain deadzone. Either the saturated position or angle increment is fed to the jogger that calculates the inverse kinematics (IK) and Jacobian to teratively increment the robots position or orientation while the input is non-zero.

**Figure 8 sensors-21-01798-f008:**
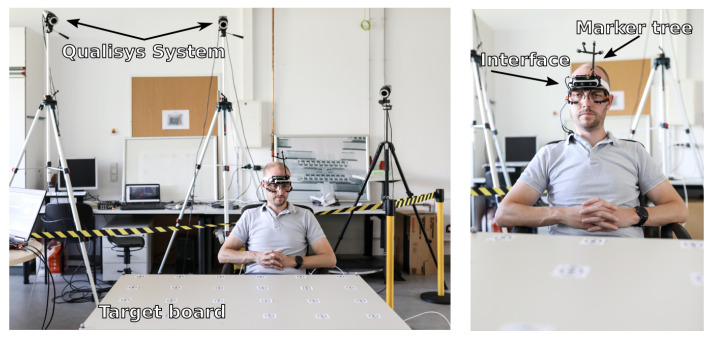
Experimental Setup. The user is wearing the interface and is sitting at a table in front of a Qualisys motion capture system. The user points at targets by either head-gaze or eye-gaze. The interface is equipped with a rigid marker tree that is attached on top of the custom camera case to capture ground truth data.

**Figure 9 sensors-21-01798-f009:**
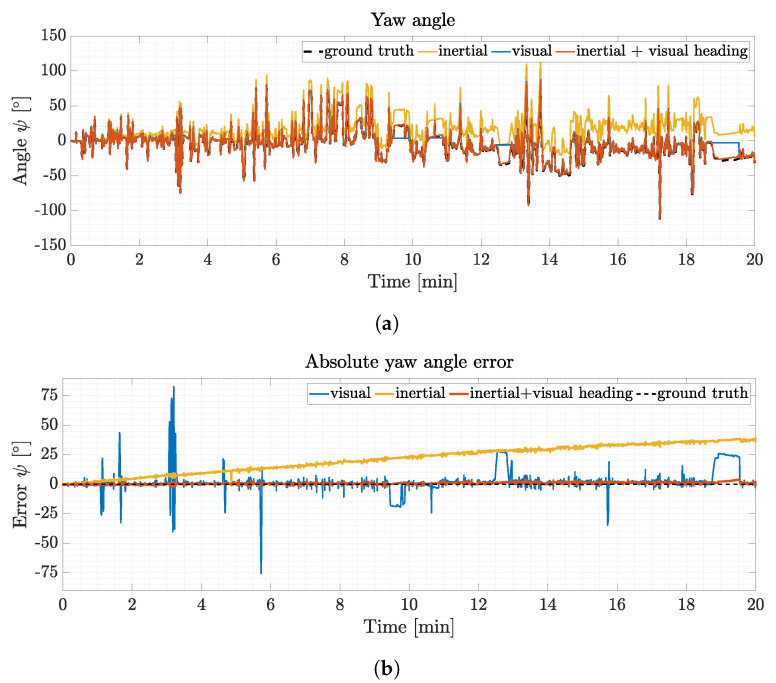
Yaw angle estimations and the corresponding absolute error for one trial: (**a**) yaw angle
comparisons between ground truth (Qualisys, black), inertial only orientation estimation (yellow),
visual SLAM orientation estimation only (blue) and the proposed orientation estimation with visual
heading vector substitute (orange). Figure (**b**) depicts the corresponding heading error referenced
to the Qualisys system for either visual only (blue), inertial only (yellow) or visual and inertial yaw
angle estimations (orange).

**Figure 10 sensors-21-01798-f010:**
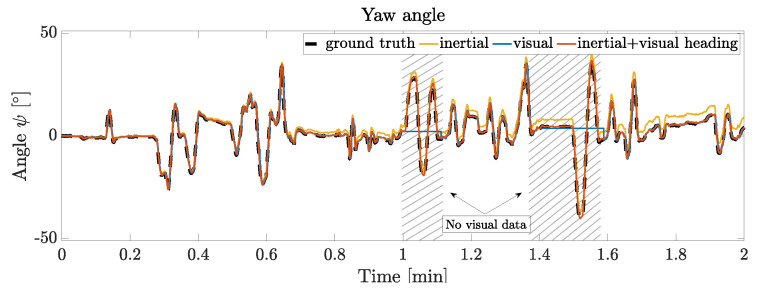
Sequence of yaw angle estimations during complete loss of visual data. Yaw Angle estimations: ground truth (black), the proposed orientation estimation (orange), the inertial only orientation estimation (yellow) and the visual orientation estimation only (blue). During complete loss of visual feedback (hatched areas) the filter switches the input heading source to the IMU heading vector to calculate reliable data until visual data is available again.

**Figure 11 sensors-21-01798-f011:**
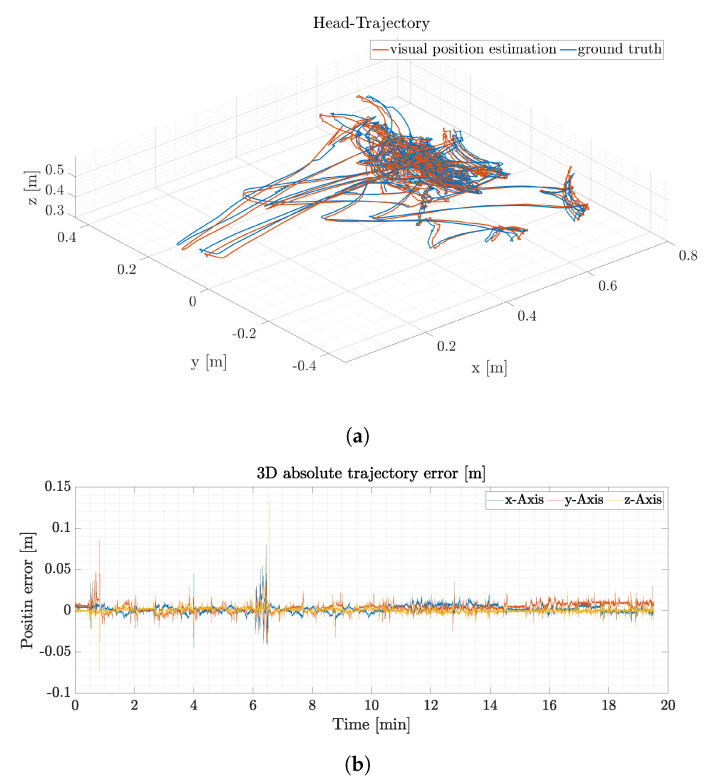
Results for head trajectory estimations for one trial: (**a**) Ground truth head trajectory
measured with the Qualisys system (blue) and the position estimation based on visual position
estimation (orange). Figure (**b**) depicts the corresponding absolute trajectory differences for each
individual axis between ground truth Qualisys data and the visual position estimation. The maximum
single axis error is 130 mm in the *z*-axis.

**Figure 12 sensors-21-01798-f012:**
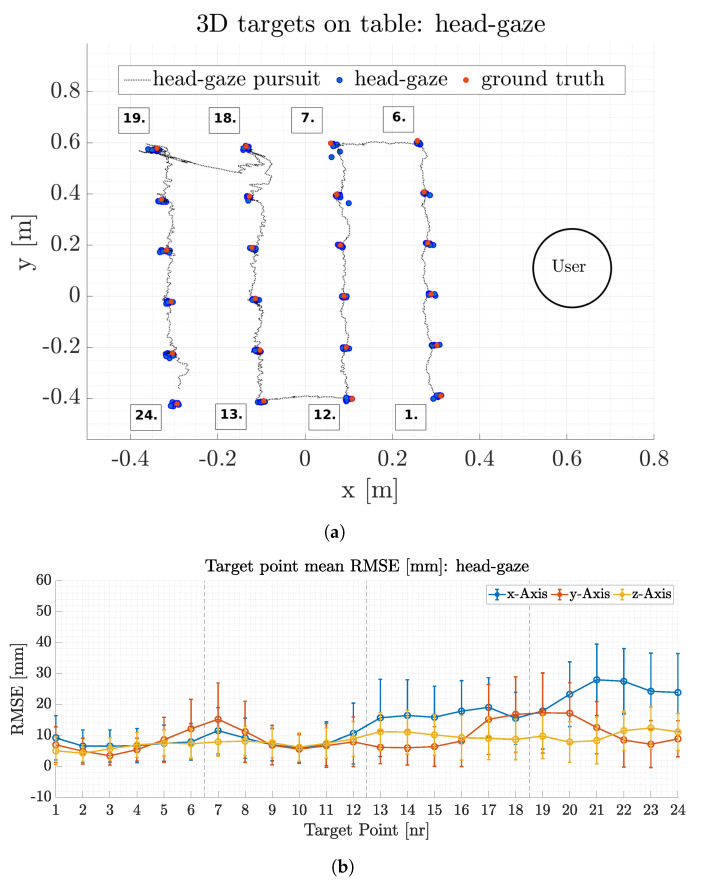
Measurement results for head-gaze accuracy: (**a**) Absolute position of head-gaze on target
position. The ground truth position of the targets (red circles) is based on pre-recorded Qualisys
measurements. The head-gaze positions are depicted as blue circles. The head-gaze trajectory
(switching between targets) is marked with a dotted black line. The users approximate position is
represented as a black circle. Figure (**b**) depicts mean head-gaze error for each individual axis along
target points throughout all trials. The black dashed lines indicate separation of the target groups for
all four rows.

**Figure 13 sensors-21-01798-f013:**
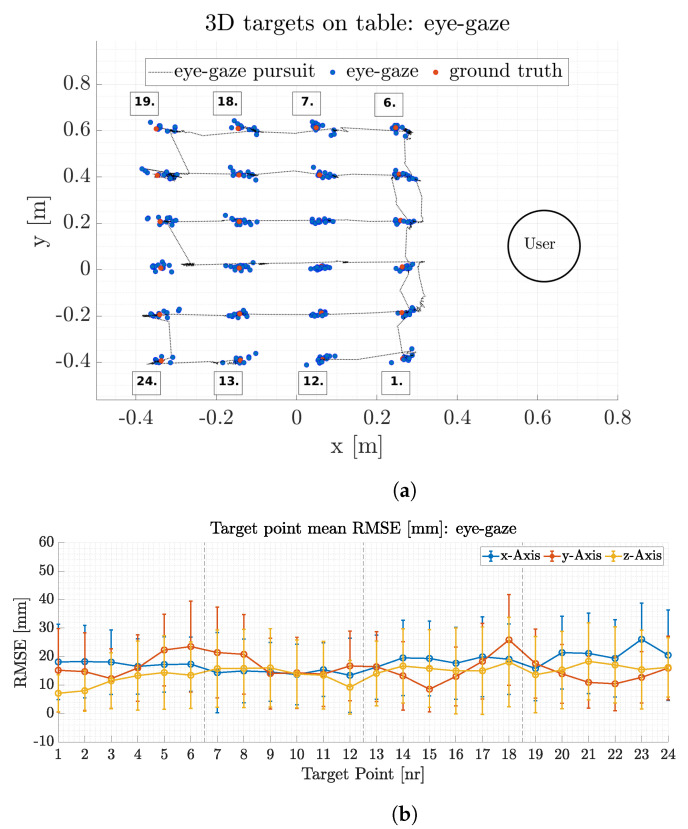
Typical measurement results for eye-gaze accuracy: (**a**) Absolute position of head-gaze on
target position. The ground truth position of the targets (red circles) is based on pre-recorded Qualisys
measurements. Absolute eye-gaze position on a target are depicted as blue circles. A sequence of one
complete transition between targets through eye-gaze is depicted as a dotted black line. The users
approximate position is represented as a black circle. Figure (**b**) depicts the mean error of the eye-gaze
for each individual axis along target points throughout all trials. The black dashed lines indicate
separation of the target groups for all four rows.

**Figure 14 sensors-21-01798-f014:**
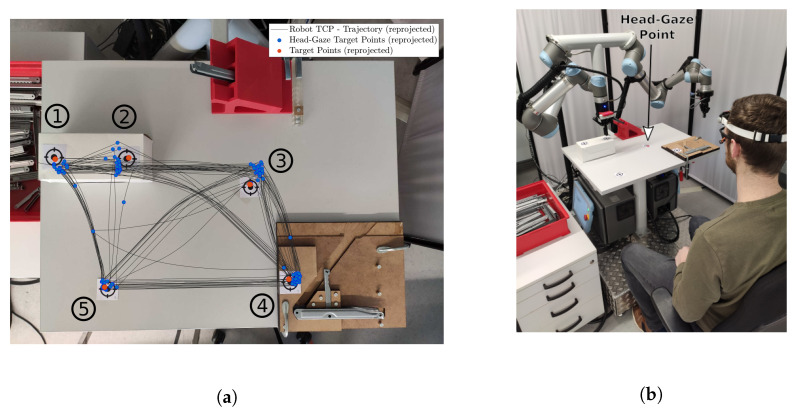
3D gaze point robot control application. (**a**) depicts the reprojection of the gaze points
(blue circles), target points (red circles) and the robots TCP (black line) onto the actual workspace
image. (**b**) shows the robot workspace application. A user is sitting in front of the dual arm UR5
robotic system. The user aims for the targets using the head-gaze approach. Upon an eye blink (left
eye), the gaze-point is transferred to the robot control pipeline.

**Figure 15 sensors-21-01798-f015:**
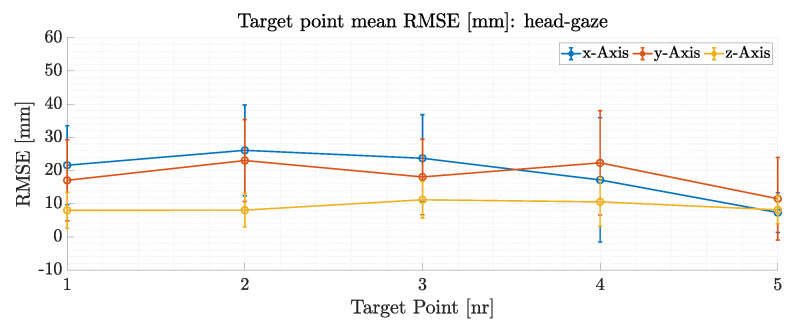
Mean RMSE error of the head-gaze for each individual axis along target points throughout five trials during robotic teleoperation.

**Table 1 sensors-21-01798-t001:** Mean of RMSE values for inertial and the proposed visual-inertial orientation estimation [Mean ± standard deviation].

	Roll [${}^{\circ }$]	Pitch [${}^{\circ }$]	Yaw [${}^{\circ }$]
visual-inertial	0.76±0.27	0.97±0.48	0.81±0.44
inertial	0.76±0.27	0.97±0.48	12.49±8.48

**Table 2 sensors-21-01798-t002:** Mean of RMSE values for visual position estimation [Mean ± standard deviation].

	x [mm]	y [mm]	z [mm]	Total [mm]
Visual pos.	16.8±18.6	20.8±21.0	8.2±5.2	28.0±28.5

**Table 3 sensors-21-01798-t003:** Mean of RMSE values for gaze position estimations [Mean ± standard deviation].

	x [mm]	y [mm]	z [mm]	Total [mm]
Head-Gaze	14.2±11.4	9.4±9.0	8.5±6.0	19.0±15.7
Eye-Gaze	17.7±12.3	15.4±12.7	14.2±12.7	27.4±21.8

**Table 4 sensors-21-01798-t004:** Motion Parameters for the UR-5 teleoperation (vel. = velocity, acc. = acceleration).

Linear Vel. [mm/s]	Rotational Vel. [rad/s]	Linear Acc. [mm/s${}^{2}$]	Joint Vel. [rad/s]	Joint Acc. [rad/s${}^{2}$]
50	0.4	100	0.4	0.7

## Data Availability

Publicly available datasets were analyzed in this study. This data can be found here: https://github.com/Gutesnacht/Head-and-Eye-Gaze-Robot-Control.git.

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
