# Peer review of "Towards Robust Robot Control in Cartesian Space Using an Infrastructureless Head- and Eye-Gaze Interface"

_sensors, 2021, doi:10.3390/s21051798_

Round 1

Reviewer 1 Report

The paper presents a novel interface for robust and real-time control of a robotic manipulator using head motion and eye-gaze. The paper is interesting, well written and properly organized. The following remarks should be addressed before publication.

The robotic system (robot UR5) used for the experimental tests is not described in the text. It is not clear if one UR5 robot or two are used in the experiment, since in Figure 14(b) two manipulators are shown.

The motion parameters of the robot used in the experimental tests should be introduced in the text (as for instance maximum velocity and acceleration values). It would be interesting to briefly discuss the security features applied for a safe human-robot collaboration.

The terms “Cartesian robot control” could be confusing since it is not clear if you mean “control of a robot in Cartesian space” or “control of a Cartesian robot”.

It would be interesting to briefly discuss the applicability of the interface to other robotic tasks, such as assembly or painting.

I suggest the following references to improve the literature review on robotic teleoperation using head motions and eye-gaze data.

Scalera, L., Seriani, S., Gasparetto, A., Gallina, P. A Novel Robotic System for Painting with Eyes. Mechanisms and Machine Science, 2021, 91, pp. 191–199.

Dziemian, S., Abbott, W. W., Faisal, A. A. (2016, June). Gaze-based teleprosthetic enables intuitive continuous control of complex robot arm use: Writing & drawing. In 2016 6th IEEE Int. Conf. on Biomedical Robotics and Biomechatronics (BioRob) (pp. 1277-1282).

Author Response

Thank you very much for the review and comments!

We highly appreciate the feedback and have hopefully addressed all remarks.

Best regards,

Lukas Wöhle

Reviewer 2 Report

The authors introduce a sensor fusion approach to solve heading and eye gaze problem with MARG and pupil-core develop suite. The method is intuitively reasonable, and the experiments are convincing. Saying if the gaze-estimation approach is as robust as the authors claimed, it would benefit a good number of relevant applications. Unfortunately, the pupil-core suite is no longer supporting real-sense camera, as the official website claimed. (To author, do you know the reason?), which makes the duplication of the introduced work extremely difficult.

To conclude, although scientifical methods are not new, and the works heavily relies on the ORB-SLAM2 as well as pupil-core frameworks. This is still a good paper due to the extensive exploration to gaze-estimation with restricted sensor capabilities.

A few concerns to the authors.

  1. The alignment / synchronization between MARG readings and the RGB-D frames within ORBSLAM2 framework is not clear.
  2. According to the experiments, the introduced method is reliable to provide reasonable heading and eye-gaze estimation. However, the comparation between the introduced method and the state-of-art works is preferred.
  3. To the best of my understanding, the commercial level MARG sensors are capable to provide services with similar performance after geomagnetism calibration. Some of them are using intrinsic Kalman filters as well. Thus, a comparison is expected.
  4. I would suggest the authors open source their code for the society, besides, video clips of the sample experiments are preferred.

Author Response

Thank you very much for the review and comments!

We highly appreciate the feedback and have hopefully addressed all concers in sufficient detail.

Best regards,

Lukas Wöhle

Round 2

Reviewer 1 Report

The paper was improved as suggested, and it can be accepted for publication in the present form.